

# Multimodal multi-objective optimization algorithm based on hierarchical environment selection strategy

Xiao Wang[1], Dan Wang[2] and Jincheng Zhou[3]

[1] State Key Laboratory of Public Big Data, College of Computer Science and Technology, Guizhou University, GuiYang, China
[2] Qiannan Normal University for Nationalities, School of Mathematics and Statistics, Duyun, China
[3] Qiannan Normal University for Nationalities, Key Laboratory of Complex Systems and Intelligent Optimization of Guizhou Province, School of Computer and Information, Duyun, China

## ABSTRACT

The article proposes an optimization algorithm using a hierarchical environment selection strategyto solve the deficiencies of current multimodal multi-objective optimization algorithms in obtaining the completeness and convergence of Pareto optimal Sets (PSs). Firstly, the algorithm in this article is framed by a differential evolutionary algorithm (DE) and uses a special crowding distance to design a neighborhood-based individual variation strategy, which also ensures the diversity, and then special crowding distance is used to help populations with non-dominated sorting. In the stage of environmental selection, a strategy of hierarchical selection of individuals was designed, which selects sorted non-dominant ranked individual layer by layer according to the ratio, which allows potential individuals tobe explored. Finally, in the stage of evolution of individuals, the convergence and diversity of populations were investigated, anddifferent mutation strategies were selectedaccording to the characteristics of individuals. DE reproduction strategies are used for iteration, preventing individuals from avoiding premature convergence and ensuring the algorithm's searchability. These strategies help the algorithm to obtain more diverse and uniformly distributed PSs and Pareto Front (PF). The algorithm of this article compares with several other excellent algorithms on 13 test problems, and the test results show that all the algorithms of this article exhibit superior performance.

Corresponding author
Dan Wang, wdan81@163.com

## INTRODUCTION

Multi-objective optimization problems (MOPs) refer to having a lot mutually exclusive multifactor in the optimization process, and solving such problems requires discovering the best solution, which not only needs a best solution but also needs a lot equivalent solutions, the essence while retaining the MOPs of searching for Pareto Frontier (PF) in solving Multimodal multi-objective optimization problems (MMOPs), researchers also need to find the Pareto front corresponding to multiple Pareto solution sets. Such as problems with the procurement of equipment (*Zhang et al., 2019*) unbalanced classification of credit card

fraud detection problems (*Han et al., 2022*), problems of path planning (*Yue, Qu & Liang, 2018*), and problems in designing rocket engines (*Kudo, Yoshikawa & Furuhashi, 2011*).

The steps for solving MOPs are divided into three steps: firstly, the related optimization problem is abstracted into a mathematical model; then, the constraints, variables, and objective functions are determined, which are classified as linear or nonlinear, discrete or continuous; finally, a suitable method is found to solve it. The relevant definitions of minimization are as follows:

$$\begin{cases} \min F(X) = \{f_1(X), f_2(X), \ldots, f_m(X)\} \\ s.t. X = (x_1, x_2, \ldots, x_n) \end{cases} \tag{1}$$

In Eq. (1), $F(x)$ is the objective function mapping the decision vector $X$ of the decision space to the objective space. min is the minimum values of the solution function respectively. $X = (x_1, x_2, \ldots, x_n)$ denotes the decision vector, which has $n$ dimensions. $m$ denotes the dimension of the objective space.

Several concepts of MOPs are as follows (*Deb, 1999*):

Theorem 1: Suppose that $X_1$ and $X_2$ are two feasible solutions to some problem, if $\forall k \in \{1, 2, \ldots, m\}, f_k(X_1) \leq f_k(X_2), \exists i \in \{1, 2, \ldots, m\}, f_k(X_1) < f_k(X_2)$, then $x_1$ is called to dominate $x_2$, and vice versa for a non-dominated solution.

Theorem 2: A non-dominated solution is Pareto Solution (PS), All non-dominated solutions together consist of the Pareto optimal Sets (PSs).

Theorem 3: The front outside formed by mapping set of PS, the surface formed is the Pareto optimal Front (PF).

Many different variants of multi-objective optimization algorithms are proposed to solve the MMOPs, from the initial pair of algorithms with non-dominated sorting genetic algorithms and decomposition-based multi-objective evolution, based on which many variants of multi-objective evolutionary algorithms are also proposed, but these variants are still ineffective. An optimization algorithm was proposed to solve MMOPs (*Deb & Tiwari, 2005*). *Liang, Yue & Qu (2016)* analyzed the challenges faced such problems and proposed a niche algorithm with decision space to solve these problems. *Yue, Qu & Liang (2018)* proposed an improved algorithm based on particle swarm that uses a ring topology and special congestion distances, results of the experiment illustrate the article finds more Pareto solutions and makes the Pareto solutions uniformly distributed. *Liu, Yen & Gong (2018)* designed a two-archiving. The two-archiving coevolution strategy improves the diversity, and the methods can get more Pareto solutions. *Tanabe & Ishibuchi (2020)* proposed an idea for solving MMOPs, which improves this algorithm's performance by assigning each child to a different sub-problem in each iteration to handle multiple equivalent solutions and comparing them with neighboring children. *Qu et al. (2020)* proposed an algorithm based on particle swarm, which uses a self-organization strategy to decompose the populations and optimize them separately, combined with a parallel search strategy to lift the diversity. *Liu et al. (2020)* proposed the convergence penalty density strategy, which first evaluates the degree of local convergence for PSs and then combines the dominance relationship between the Euclidean distance to get a new distance, and finally uses this distance to estimate the density value as a selection criterion. *Javadi, Zille*

*& Mostaghim (2019)* analyzed that some algorithms are a drawback in solving MMOPs in response to some algorithms that are closely based on the objective function values as a selection criterion in the environment selection, so a new congestion distance operator and neighborhood variation operator are designed to solve MMOPs. *Fan & Yan (2021)* proposed a partitioned strategy, and have broken the decision space down into several parts; this method ensures the diversity, and eventually diminishes the difficulty in the search. *Li et al. (2019)* designed a reinforcement learning DE algorithm, which leads the population evolution and then find multiple solutions with fitness ranking. *Liang et al. (2018)* used a self-organization mechanism to find the population distribution structure and establish a neighborhood search while using a special non-dominated ranking method. *Yue et al. (2021)* proposed an improved evolutionary optimization algorithm with crowding distance, the algorithm first calculates the degree of crowding between two spatial individuals and takes the individual with a higher degree of crowding as the current offspring. Although the algorithm improves the variety of solutions to some extent, the algorithm still has an incomplete set of Pareto solutions. *Li et al. (2022)* proposed an algorithm with hierarchical rank to maintain the diversity of populations. *Liang et al. (2022a)* and *Liang et al. (2022b)* proposed an exceptional environment selection method to solve MMOPs. *Yan et al. (2022)* proposed a DE algorithm with metrics and Pareto hierarchy. *Qu et al. (2022)* proposed a method, which makes enhancement of the search capability of the algorithm using a network technique to split the decision space into a lot grids. Although the algorithm enhances the diversity to some extent, the algorithm still needs an incomplete set of PSs. *Liang et al. (2023)* proposed a neighborhood search strategy with a data interpolation technique, and the algorithm can help the population evolve better and enhance its search capability. *Liang et al. (2022a)* and *Liang et al. (2022b)* proposed a particular environmental selection strategy to enhance population diversity. This algorithm obtains feasible solutions and retains more well-distributed PSs. *Li et al. (2022)* proposes a hierarchical ranking scheme and a method for assessing the quality of local convergence to preserve the diversity better. *Ji, Wu & Yang (2024)* proposes a population conservation algorithm to identify different PSs in known areas while explore new areas using DE. *Sun, Chang & Zheng (2021)* proposed a evolutionary algorithm using a ring topology structure that can adapt to changes. *Zhang et al. (2021)* proposed a two-stage dual small-habitat evolutionary strategy. Solutions to the MMOPs are broken down into two parts, which use the small-habitat strategy in the decision space firstly. Secondly, the author uses the small-habitat strategy in two spaces.

Solving MMOPs requires finding Pareto fronts with approximation and diversity, and finding a sufficient number of equivalent solutions. The algorithms mentioned above improve the performance of the algorithms in the decision space by designing a series of search strategies to find multiple PSs and increase the diversity of PSs, thus obtaining more complete and evenly distributed PSs and PF. However, these algorithms are still flawed in their ability to search for PSs, and their searches yield insufficiently diverse and poorly distributed PSs, while the algorithms described above are not fully consider the performance of the algorithms in the objective space, such insufficient and insufficiently capable algorithms for searching PSs together lead to insufficient diversity and poor

distribution of PSs and PF. In this article, a multi-objective optimization algorithm (MMODE_ES) is proposed to address these problems.

In this article, firstly, the mutation phase and environment selection phase of the differential evolutionary algorithm is improved, the base vector is selected by combining special crowding distances of individuals, and the mutation strategy is designed to improve the solution performance of the algorithm; secondly, the hierarchical environment selection scheme is designed to retain the potential individuals and improve the exploration capability of DE; finally, the diversity and convergence of individuals in the process of evolution are investigated, and the diversity and convergence of individuals in the process of evolution is further of improving the convergence and diversity of PSs and PF. In the end, the experiments show that the strategy of MMODE_ES works.

The next relevant work is described as follows: the relevant definitions in the MMOPs and the methodologies used for the difficulties are present in "Related Work". "Methods" provides a detailed introduction to the proposed method. The "Experiment" discusses the experimental methodology, compares the algorithms, and analysis in the results. "Conclusion" provides a summary.

## RELATED WORK

This section first gives an introduction to MMOPs, then describes the evaluation metrics for algorithms that address MMOPs, and finally describes the algorithmic framework and related techniques used in this article to address MMOPs

### Definitions for MMOPs

MMOPs are ubiquitous in real life, and these types of problems are also a special type of MOPs. When defining MMOPs, the following situations must occur:

1. Firstly, there are some global PSs;

2. secondly, scenario is that there is at least one local PS in the decision space.

However, the connection between these PSs and the objective values in the objective space is not a one-to-one mapping, but rather multiple PSs corresponding to a single objective value. Compared with general MOPs, algorithms need to spend more resources and time searching for more different optimal equivalent solutions when solving MMOPs.

### Evaluation indexes for MMOPs

The traditional evaluation index of the MOPs only focuses on the performance of the population; the evaluation index of multimodal multi-objective optimization (MMO) optimization also needs to focus on its decision space. Common metrics for MMO algorithms are Pareto solutions proximity (*PSP*) (*Liang et al., 2018*), Hypervolume (*HV*) (*Zitzler & Thiele, 1999*), and Inverted Generational Distance (*IGDX*) (*Zhou, Zhang & Jin, 2009*), *IGDX* index can measure the performance of the PS; the *PSP* index can reflect both the convergence and diversity of the solution set obtained by the algorithm and can also measure the coverage of the PS on the real PS, *PSP* is more comprehensive and reasonable evaluation. The convergence and diversity of *HV* and PF are directly proportional. In the experimental stage, this article uses the *PSP* and *HV* metrics to evaluate the effectiveness of MMO algorithm, the equations for *HV* and *PSP* are as follows:

$$PSP = \frac{CR}{IGDX} \tag{2}$$

$$IGDX\left(PS, PS^*\right) = \frac{\sum_{a \in PS} d(a, PS^*)}{|PS|} \tag{3}$$

$$HV\left(PS, P\right) = volume \bigcup_{x \in PF} v(x, P) \tag{4}$$

In the above equation, $CR$ is the coverage rate of the PSs, where $P$ is the reference point, which represents the hyperbolic volume surrounded by the solutions and the reference point. The $PSP$ measures the diversity and convergence and the larger $PSP$, the more equivalent solutions the algorithm obtains. The effect shown by the algorithm is proportional to the $HV$.

## Differential evolution

Differential evolution algorithm (DE) is an efficient and straightforward evolution algorithm proposed (*Das & Suganthan, 2011*; *Qin, Huang & Suganthan, 2009*). Among many heuristic algorithms, the advantages of DE are more obvious. Firstly, compared to others, DE has a shorter running time; secondly, the algorithm framework of DE is simpler, the algorithm process is easy to understand, and researchers can quickly and deeply understand it; finally, DE converges faster compared to other evolutionary algorithms. Therefore, DE is more effective in many standard test functions and real-world problems. The process is consistent with many similar algorithms, and DE has four processes: firstly, randomly generating a population; using mutation equations to induce population variation; crossing populations and selecting suitable individuals; finally, environmental selection is the process of selecting outstanding individuals to form offspring populations.

## Specialized crowding distances

Crowding distances ($CD$) is to measure the degree of crowding of individuals in a population, for calculating the crowding distances of individuals, and the algorithms use crowding distance metrics to enhance the population's diversity (*Ghorbanpour, Jin & Han, 2022*). In MMOPs, the algorithm uses special crowding distances ($SCD$) to measure the degree of crowding of population individuals (*Yue et al., 2021*). The non-dominated sorting method using special crowding distances can improve the performance and efficiency of MO algorithms. The MMODE_ES uses the special crowding distance in the non-dominated sorting of individuals to solve the problem of selecting individuals from a dense set of solutions.

## METHODS

In this section, MMODE_ES details and processes are described.

## Differential variation strategies

Many different differential mutation strategies are designed according to other optimization problems give the searchability of the DE a boost. In this article, the algorithm adopts the differential variation strategies of DE/best/2 and DE/rand/2 to iterate.

To improve the searchability of the differential evolution algorithm, many different differential mutation strategies are designed according to other optimization problems, and the common ones are DE/rand/1, DE/best/1, DE/best/2, and DE/rand/2. In this article, the algorithm adopts the differential variation strategies of DE/best/2 and DE/rand/2 to iterate. The equation for DE/best/2 strategy is as follows:

$$V_i = X_{bset} + F \cdot \left[ \left( X_{r_1} - X_{r_2} \right) + \left( X_{r_3} - X_{r_4} \right) \right] \tag{5}$$

In this equation, $X_{best}$ is the optimal individual in the current population, $X_{r1}$, $X_{r2}$, $X_{r3}$ and $X_{r4}$ are random individual in the population.

The equation for DE/rand/2 strategy is as follows:

$$V_i = X_{r_1} + F \cdot \left[ \left( X_{r_2} - X_{r_3} \right) + \left( X_{r_4} - X_{r_5} \right) \right] \tag{6}$$

In this equation, $X_{r1}$, $X_{r2}$, $X_{r3}$, $X_{r4}$, and $X_{r5}$ are random individuals.

The DE/rand/2 strategy enhances the diversity of the population because it randomly selects individuals for evolution so that the algorithm can explore more extensively in the search space. DE/best/2 focuses more on global search because it uses the best-performing individuals as a reference, which drives the algorithm to search for more promising regions, which facilitates the improvement of the convergence of the population.

The algorithm uses the DE/rand/2 strategy at the beginning of the evolution to ensure the diversity of the population obtaining more evenly distributed PSs. In the subsequent iterations, after the population is non-dominated sorted, the variety and convergence of the population need to be investigated at this time; if the current non-dominated rank is only 1 level, and the algorithm tends to converge at this time, it is necessary to improve the diversity of the population. In the next generation, the DE/rand/2 strategy can help population to evolve. Suppose there are multiple levels of population non-dominance rank. In that case, it is necessary to explore PSs closer to the actual PSs to improve the algorithm's convergence, and DE/best/2 strategy can help the population's evolution.

## Neighborhood variation strategy

The primary process of the DE is the differential strategy; the algorithm selects appropriate individuals for mutation, which can ensure the population's diversity and distribution and improve the algorithm's searchability. The traditional differential mutation strategy randomly selects five individuals in the current population for cross-mutation, and the random selection of individuals cannot guarantee the population's diversity and the algorithm's convergence. To ensure the diversity and the variety, this article proposes a neighborhood variation strategy.

Neighborhood variation strategy adopts the idea of small populations. Select the congestion distance that is large in the first 20 individuals to form a small population according to the crowding distance of the current individual, and in the small population

to choose the most considerable congestion distance of an individual as the base vector, and then finally randomly select individuals, and ultimately enter the mutation stage, which can improve the diversity. Based on the crowding distance for objective space in the current individual, form these population of the top 20 individuals with larger crowded distances. Using the individual with the largest crowding distance in these population as the basis vector, and then randomly selected individuals are used, and ultimately enter the mutation stage, which can improve the diversity in the objective space.

$CD$ is calculated by weighted Euclidean average distance to individuals in the neighborhood in decision space. The equation for its calculation:

$$CD_{i,x} = \sum_{j=1}^{k_1} (k_1 - j + 1) d_{i,j} \tag{7}$$

$d_{i,j}$ indicated the Euclidean distance, which is the distance between $i^{th}$ and $j^{th}$. The special congestion distance ($SCD$) equation is as follows:

$$SCD = \begin{cases} max\left(CD_{i,x}, \dfrac{CD_{i,F}}{Rank}\right), CD_{i,x} > CD_{agv,x} \ or \ CD_{i,f} > CD_{agv,f} \\ min\left(CD_{i,x}, CD_{i,f}\right), otherwise \end{cases} \tag{8}$$

In Eq. (8), $CD_{avg,x}$ denotes the average crowding distance, $CD_{i,f}$ denotes the crowding distance of an individual, $CD_{avg,f}$ denotes the average crowding distance, and $Rank$ denotes the current individual's nondominant rank.

First of all, when the population starts to evolve, the mutation strategy randomly selected one individual as the base vector, and randomly select the individual with a higher probability in the whole population to improve the exploration ability and prevent premature algorithm stagnation. To improve search efficiency, researchers not only need to get the uniformly distributed Pareto solution set but also need to get the uniformly distributed Pareto frontier surface; the mutation strategy needs to consider the distributivity of the population in both spaces at the same time, simultaneously, the population should be two mutation strategies guiding the evolution.

The reason for adopting the above strategy is the adaptive balance of exploration and exploitation capacity, where both decision-making and objective spatial diversity are improved. In the later stages of evolution, MMODE_ES will select the neighbors to improve search capabilities. Selecting adjacent individuals for evolution in two different spaces can enhance diversity in various spaces. MMODE_ES framework is as follows:

## Layered environment selection strategy

To further improve performance on the objective space so that MMODE_ES obtains a more uniformly distributed PSs, the article proposes a hierarchical environment selection strategy, which selects the nondominated solutions of each layer in different proportions in different evolutionary stages. In the pre-evolutionary step, the algorithm selects suitable individuals for the next phase of evolution in a certain proportion of the population after non-dominated sorting with the special crowding distance, and individuals with high

---

**Algorithm 1: Mutation Strategies**

---

Input: population size $N$, initialization
population $P$, Neighborhood size $k$
Output: population $POP$

Evaluate $P$
Calculation of individual $CD$

for $i = 1:N$

Select $k$ individuals neighboring $i$
$M_{population} = k$ individuals in decision space
$N_{population} = k$ individuals in objective space
Select individuals randomly from $M$ and $N$
$i_1$ by $M$, $i_2$ by $N$, and $i_3$ by $P$ according to Eqs. (5) and (6)
$POP = POP \cup i_1 \cup i_2 \cup i_3$
end if

---

non-dominated rank and large special crowding distances have a higher probability of being selected. In the early stage of individual evolution, selecting individuals proportionally, instead of selecting all individuals with high dominance rank, can prevent the algorithm from converging prematurely and give the individuals with lower non-dominance rank in the early stage a chance to explore to obtain more Pareto solution sets, and at the same time, let the algorithm no longer spend more time on inefficient searching for the individuals with lower non-dominance rank, and in the early phase, the selection of vast majority of individuals are those in the first three nondominant ranks. The individuals in the first rank are greater than the number of individuals in the second rank and more significant than those in the third rank. In the late phase, the algorithm gradually converges, and the Pareto solution set and Pareto frontier surface progressively perfect. At this time, the algorithm enters the second search stage, where the environment selects only the individuals with higher non-dominated ranks.

The proportion $Ra$, $Rb$, $Rc$ of hierarchical selection is calculated as follows:

$$Ra = \begin{cases} 0.6 + 0.8 \cdot (gc - 1), 1 < gc < G \\ 1, G < gc < Maxgen \end{cases} \quad (9)$$

$$Rb = 0.8 \cdot Ra, Ra < 1 \quad (10)$$

$$Rc = 0.6 \cdot Ra, Ra < 1 \quad (11)$$

In the evolution of the population, the top non-dominated ranked individuals are not necessarily optimal, and the proportional selection of the top-ranked individuals is to remove the pre-evolutionary high non-dominated ranked but undesirable solutions and to give the low non-dominated ranked but potential individuals a chance to be explored. From Eqs. (9) to (11), we can see that $Ra > Rb > Rc$, so the hierarchical selection of individuals is mainly selected from the first three levels of individuals; the selection of individuals with

a high non-dominated rank tends to allow the algorithm to search toward the potential region, but only selecting the first level of individuals as a child will make the algorithm fall into the local optimum prematurely. The selection of individuals at each level will allow the algorithm to search too much for the individuals who do not have the potential. The algorithm will require more work to converge. Therefore, the algorithm mainly selects the top three individuals in the non-dominated rank and selects the low-rank individuals with small probability, which can help the algorithm to search towards the potential area and also give the lower level of individuals a chance to be searched, which not only improves the diversity of the population but also improves the closeness of the population to the actual population. The framework of the algorithm is as follows:

---

**Algorithm 2: Herarchical Environmental Selection Strategy**

---

Input: population size *N*, population *POP*,
Non-dominated hierarchical layers, *frontmax*
Output: population $POP_1$

Calculate special crowding distance
Non-dominated sorting for *POP*
for 1: *frontmax*
Select the individuals according to Eqs. (9) to (11)
*PO* = Combination individuals
$POP_N$ = selected N from the *PO*
end for

---

## Framework of MMODE_ES

The algorithm in this article is called MMODE_ES, it has the following steps:

---

**Algorithm 3: MMODE_ES**

---

Input: Population size, Max generations, Max evaluations
Output: Population *P*

Initialization of population *P*
Evaluate *P*
Calculate population *P* crowding
**while** satisfaction of termination conditions **do**
Select differential variants
Cross-variation according to Eqs. (5) and (6)
Population *O* by combining
Evaluate *O*
Calculate population *O* crowding
Combining *O* and *P* for non-dominated sorting
Environmental selection according to Eqs. (9) to (11)
**end while**

---

# EXPERIMENT

This section provides experimental validation of the proposed algorithm. It describes the relevant parameter settings and the test set used for the experiment and finally analyzes and summarizes on the experimental results.

**Table 1  The parameter setting in experiments.**

| Parameter | Value |
|---|---|
| Maximum fitness | 10,000 |
| Maximum number of generations | 50 |
| F | 0.5 |
| CR | 0.5 |
| Population size | 100*$Nvar$ |

## Experimental settings

All algorithm parameters are set according to the original author's suggestions, so that the algorithm can be proved to be efficient, which can allow individual algorithms to be compared at one level. The algorithm stops running when the fitness reaches its maximum, the setting of the relevant parameters is related to dimensions of the decision vector ($Nvar$) in test problems, the maximum fitness is set to 10000 and the population size to 100*$Nvar$. The maximum number of generations is set to 50. The $CR$ is 0.5 and the $F$ is 0.5. The relevant settings are described in Table 1. The running environment of this experiment is Windows 11 operating system, AMD Ryzen 7 4800H with Radeon Graphics 2.90 GHz and 16.0 GB memory. The software uses Matlab2021.

## Comparison of algorithms

To illustrate MMODE_ES's effectiveness in this article, five excellent MMO algorithms are selected for comparison test, which are MO_Ring_PSO_SCD ((*Yue, Qu & Liang, 2018*), DN-NSGAII (*Liang, Yue & Qu, 2016*), Omni-optimizer (*Deb & Tiwari, 2005*), TriMOEA-TA&R (*Liu, Yen & Gong, 2018*), MMODE_ICD (*Yue et al., 2021*) these algorithms. MO_Ring_PSO_SCD and TriMOEA-TA&R are often MMO algorithms that are compared and used as improvements by many researchers. Omni-optimizer and DN-NSGAII are classical MMO algorithms. The MO_Ring_PSO_SCD algorithm is the best algorithm in the CEC2019 multimodal optimization problem competition. MMODE_ES in this article is compared with these MMO algorithms.

## Test functions

This article uses 13 test functions on the CEC2019 test problems were selected to test the search ability of the proposed method for demonstrating whether the MMODE_ES in this article is feasible, including functions: MMF1 to MMF10 (*Yue et al., 2019*); SYM-PART simple and SYM-PART simple rotated (*Rudolph, Naujoks & Preuss, 2007*); omni-test (*Deb & Tiwari, 2008*). These benchmark test problems contain Pareto fronts of different shapes and Pareto solution sets. Some of the real PSs and PF in these testing questions have complex situations, with overlapping, numerous, multi-dimensional, and complex shapes that pose great challenges to MMODE_ES.

## Experimental results and analysis

The *PSP* metric can evaluate search ability, in other words, it can measure the degree of closeness between the obtained PSs and the actual PSs. *HV* can measure the convergence and diversity in the objective space. The article uses *rPSP* and *rHV* to facilitate the analysis.

*rPSP* is the inverse of *PSP*, and *rHV* is the inverse of *HV*. Therefore, the smaller these two metrics are, the better the performance. MMODE_ES and several other comparative algorithms were tested 30 times on 13 test functions. This article summarizes the results of 30 tests and calculates the mean and standard deviation. In the experiment, the Wilson test was used to detect whether there was a significant difference in the results. The symbol "+" in the table indicates that MMODE-ES performs better than the comparison algorithm. Conversely, the symbol "-" indicates that MMODE_ES performs worse than the comparison algorithm, while the symbol " =" indicates that there is no difference in performance between MMOD_ES and the comparison algorithm. Meanwhile, the Mean in the table represents the average of 30 results, which can indirectly reflect the overall level of each algorithm, while Std represents the standard deviation of these 30 results, which indirectly represents the stability of the algorithm. Therefore, the smaller the Mean and Std of the algorithm's results, the better the algorithm. Therefore, to demonstrate the excellence of MMODE_ES, some data has been marked in bold, indicating that the results of MMODE_ES are better.

As the results in Tables 2 and 3 show that the algorithm MMODE_ES in this article performs well on most of the test functions. By comparing the *rPSP* and *rHV* metrics in the table, the algorithm in this article, and comparing it with many excellent algorithms, the operation of the algorithm in this article is effective. In the *rPSP* index: except for the DN-NSGAII algorithm in the SYM-PART simple function, MMODE_ES is at a disadvantage in the rest of the test function obtained better results, which shows that MMODE_ES searched for the PSs has a better distribution in the MMOPs is more competitive. For the *rHV*: MMODE_ES gives excellent results on 13 test problems, and experiments show that this article's algorithm also obtains solutions with better convergence and distribution. Combining the results in Tables 2 and 3, MMODE_ES shows overwhelmingly satisfactory results than a lot comparative MMO algorithms, so MMODE_ES's operation in this article is effective.

The optimization results of this method are compared with several comparative algorithms on the MMF3 to show the effectiveness of this algorithm more intuitively, and the results are shown in the form of scatter plots to show the results. In MMF3, the distribution of the decision space solutions of MMODE_ES is good, with very few missing solution sets. In contrast, several other algorithms have significantly more missing solution sets than the present algorithm. The evolved distribution of the solution sets is, to a great extent, able to reach the overlap with the real solution set. In the final Pareto front, the Pareto front obtained in this article is complete and close to the real show to a large extent. The Pareto front obtained by other algorithms can also be close to the real front to a large extent. Therefore, MMODE_ES has the best performance.

The PSs for all algorithms on the MMF3 in the decision function are as Fig. 1, Fig. 1 shows the real PSs of this test problem. The results of all algorithms on the objective space over the MMF3 function are as Fig. 2, Fig. 2 shows the real results of this test problem. From the results shown on the listed test problems, MMODE_ES has better results. In this article, MMODE-ES searched for more and distributed PSs, which approaches the actual situation; the distribution of the PSs is uniform, and missing PSs are less. In the

**Table 2  Comparison results with other excellent algorithms on *rPSP* index.** PSP is the inverse of PSP. Therefore, the smaller metric is the better the performance. The algorithm runs on 13 test functions and independently runs 30 times, which gets the average and standard deviation. The data in the table are the average of the 30 times. The experiment tested the data obtained by all algorithms for significant differences using the Wilson test. The "+" and "-" in the table indicate statistically superior or inferior to the comparison algorithm, respectively, while "=" indicates statistically similar results. In Table 1, Mean represents the average of 30 results. Std represents the standard deviation on the test problem.

| | | MO_Ring_PSO_SCD | DN-NSGAII | Omni-optimizer | TriMOEA-TA&R | MMODE_ICD | MMODE_ES |
|---|---|---|---|---|---|---|---|
| MMF1 | Mean | **0.0488+** | **0.0969+** | **0.0975+** | **0.0735+** | **0.0493+** | 0.0417 |
| | ± Std | **0.0019** | **0.0145** | **0.013** | **0.0108** | **0.0030** | 0.0013 |
| MMF2 | Mean | **0.0465+** | **0.1742+** | **0.1643+** | **0.0931+** | **0.0247+** | 0.0075 |
| | ± Std | **0.0147** | **0.1368** | **0.1305** | **0.0574** | **0.0054** | 0.0010 |
| MMF3 | Mean | **0.0335+** | **0.1172+** | **0.1351+** | **0.0871+** | **0.0209+** | 0.0068 |
| | ± Std | **0.0099** | **0.0759** | **0.0994** | **0.0275** | **0.0037** | 0.0007 |
| MMF4 | Mean | **0.0273+** | **0.0771+** | **0.0840+** | **0.1537+** | **0.0257+** | 0.0217 |
| | ± Std | **0.0020** | **0.0137** | **0.0238** | **0.2262** | **0.0031** | 0.0010 |
| MMF5 | Mean | **0.0869+** | **0.1773+** | **0.1789+** | **0.1132+** | **0.0853+** | 0.0711 |
| | ± Std | **0.0060** | **0.0217** | **0.0245** | **0.0126** | **0.0039** | 0.0035 |
| MMF6 | Mean | **0.0733+** | **0.1427+** | **0.1523+** | **0.0958+** | **0.0713+** | 0.0637 |
| | ± Std | **0.0043** | **0.0150** | **0.0185** | **0.0123** | **0.0043** | 0.0004 |
| MMF7 | Mean | **0.0267+** | **0.0553+** | **0.0511+** | **0.0672+** | **0.0263+** | 0.0206 |
| | ± Std | **0.0015** | **0.0151** | **0.0127** | **0.0520** | **0.0046** | 0.0023 |
| MMF8 | Mean | **0.0678+** | **0.2799+** | **0.3149+** | **0.3974+** | **0.1303+** | 0.0504 |
| | ± Std | **0.0042** | **0.0911** | **0.1326** | **0.1572** | **0.0352** | 0.0048 |
| MMF9 | Mean | **0.0082+** | **0.0219+** | **0.0316+** | 0.0031- | 0.0047- | 0.0061 |
| | ± Std | **0.0008** | **0.0078** | **0.0269** | 0.0001 | 0.0003 | 0.0004 |
| MMF10 | Mean | 0.1708- | **1.3420+** | **3.1260+** | 0.2014= | 0.2011= | 0.2035 |
| | ± Std | 0.0231 | **2.3280** | **3.3420** | 0.0001 | 0.0010 | 0.0024 |
| SYM-PART simple | Mean | **0.1776+** | **5.4511+** | **6.4608+** | 0.0210- | 0.0427- | 0.0441 |
| | ± Std | **0.0226** | **2.576** | **3.0139** | 0.0021 | 0.0055 | 0.0042 |
| SYM-PART rotated | Mean | 0.2784- | **4.4542+** | **5.4757+** | **2.0975+** | **0.0892+** | 0.0536 |
| | ± Std | 0.2500 | **1.4104** | **3.4185** | **1.4432** | **0.0153** | 0.0033 |
| Omni-test | Mean | **0.4279+** | **1.5563+** | **1.7939+** | **0.7547+** | **0.0512+** | 0.0430 |
| | ± Std | **0.0954** | **0.2878** | **0.6207** | **0.2166** | **0.0036** | 0.0020 |
| +/=/- | | 11/0/2 | 13/0/0 | 13/0/0 | 10/1/2 | 10/1/2 | |

**Notes.**
rPSP results and rHV results obtained by these comparison algorithms are underlined and shown in bold when the rPSP results and rHV results obtained by MMODE_ES are better than the comparison algorithms under the same test function.

objective space, PF of this algorithm near to the real front and better distributed than the comparison algorithm. Combined with the tabular *rPSP* and *rHV* value this article's algorithm effectively solves MMOPs.

Overall, MMODE_ES is effective in solving MMOPs, which is attributed to the fact that the algorithm's performance in decision space and objective space are considered simultaneously in the process of searching for PSs. Firstly, the designed variation strategy endeavors to combine the study of the distributivity and diversity of PSs in the process of evolution, which improves the performance of MMODE_ES in the decision space and improves the diversity of PSs; secondly, the performance of the decision space is fully

**Table 3  Comparison results with other excellent algorithms on *rHV* index.** rHV is the inverse of HV. Therefore, the smaller metric is. the better the performance. The algorithm runs on 13 test functions and independently runs 30 times, which gets the average and standard deviation. The data in the table are the average of the 30 times. The experiment tested the data obtained by all algorithms for significant differences using the Wilson test. The "+" and "-" in the table indicate statistically superior or inferior to the comparison algorithm, respectively, while "=" indicates statistically similar results. In Tables 1 and 2, Mean represents the average of 30 results. Std represents the standard deviation on the test problem.

| | | MO_Ring _PSO_SCD | DN-NSGAII | Omni-optimizer | TriMOEA-TA&R | MMODE_ICD | MMODE_ES |
|---|---|---|---|---|---|---|---|
| MMF1 | Mean | **1.1484+** | **1.1498+** | **1.1481+** | 0.4916- | 1.1458- | 1.1467 |
| | ± Std | **0.0005** | **0.0017** | **0.0011** | 1..8164 | 0.0003 | 0.0004 |
| MMF2 | Mean | **1.1855+** | **1.1947+** | **1.1855+** | **1.1835+** | **1.1765+** | 1.1481 |
| | ± Std | **0.0125** | **0.0292** | **0.0405** | **0.0130** | **0.0062** | 0.0012 |
| MMF3 | Mean | **1.1740+** | **1.1782+** | **1.1808+** | **1.1865+** | **1.1672+** | 1.1471 |
| | ± Std | **0.0060** | **0.0203** | **0.0355** | **0.0146** | **0.0048** | 0.0006 |
| MMF4 | Mean | **1.8614+** | **1.8575+** | 1.8552= | 0.9761- | 1.8522= | 1.8556 |
| | ± Std | **0.0021** | **0.0010** | 0.0008 | 1.6436 | 0.0005 | 0.0035 |
| MMF5 | Mean | **1.1483+** | **1.1487+** | **1.1472+** | **1.1502+** | **1.1461+** | 1.1456 |
| | ± Std | **0.0004** | **0.0011** | **0.0008** | **0.0019** | **0.0003** | 0.0001 |
| MMF6 | Mean | **1.1491+** | **1.1502+** | **1.1473+** | **1.1502+** | 1.1456= | 1.1458 |
| | ± Std | **0.0014** | **0.0040** | **0.0012** | **0.0034** | 0.0002 | 0.0006 |
| MMF7 | Mean | **1.1485+** | **1.1493+** | **1.1473+** | **1.1905+** | 1.1453- | 1.1461 |
| | ± Std | **0.0008** | **0.0014** | **0.0006** | **0.0838** | 0.0002 | 0.0005 |
| MMF8 | Mean | **2.4050+** | **2.3819+** | 2.3745= | **2.3806+** | 2.3764= | 2.3750 |
| | ± Std | **0.0159** | **0.0053** | 0.0010 | **0.0020** | 0.0038 | 0.0034 |
| MMF9 | Mean | **0.1034+** | **0.1034+** | 0.1033- | **0.1047+** | 0.1032- | 0.1034 |
| | ± Std | **2.6409e−05** | **2.7824e−05** | 3.1523e−05 | **9.7715e−05** | 1.4122e−05 | 1.6431e−05 |
| MMF10 | Mean | **0.0797+** | **0.0817+** | **0.0807+** | 0.0781= | 0.0784= | 0.0782 |
| | ± Std | **0.0005** | **0.0027** | **0.0031** | 0.0001 | 0.0023 | 0.0004 |
| SYM-PART simple | Mean | **0.0605+** | 0.0601= | 0.0601= | 0.0601= | 0.0600- | 0.0601 |
| | ± Std | **5.6491e−05** | 1.1505e−05 | 6.5325e−06 | 1.5433e−05 | 6.5117e0-6 | 2.0249e−05 |
| SYM-PART rotated | Mean | **0.0606+** | **0.0601+** | **0.0601+** | **0.0602+** | **0.0601+** | 0.0601 |
| | ± Std | **8.8876e−05** | **1.2144e−05** | **5.3381e−06** | **1.4502e−05** | **5.6271e−06** | 4.6401e−06 |
| Omni-test | Mean | **0.0190+** | **0.0189+** | 0.0188- | **0.0190+** | **0.0189+** | 0.0189 |
| | ± Std | **1.7289e−05** | **4.1602e−07** | 4.7385e−07 | **1.6108e−05** | **2.9395e−06** | 1.2258e−06 |
| +/=/- | | 13/0/0 | 12/1/0 | 8/3/2 | 9/2/2 | 5/4/4 | |

**Notes.**
rPSP results and rHV results obtained by these comparison algorithms are underlined and shown in bold when the rPSP results and rHV results obtained by MMODE_ES are better than the comparison algorithms under the same test function.

considered, and such consideration improves the diversity and distributivity of PF; lastly, the hierarchical environmental selection scheme allows the excellent individuals to be retained, and such design allows the distributivity and diversity of PSs to be enhanced. After comparing with these excellent algorithms, it can be seen that although some of these algorithms make efforts to search for PSs in the decision space, these search strategies still need to be further improved; meanwhile, the traditional PS selection scheme selects individuals with high nondominant rank, which does not retain the dominant individuals well, and then leads to a part of PSs being excluded from the real PSs, and the final searched PF and PSs have a missing; at the same time part of the algorithm can consider

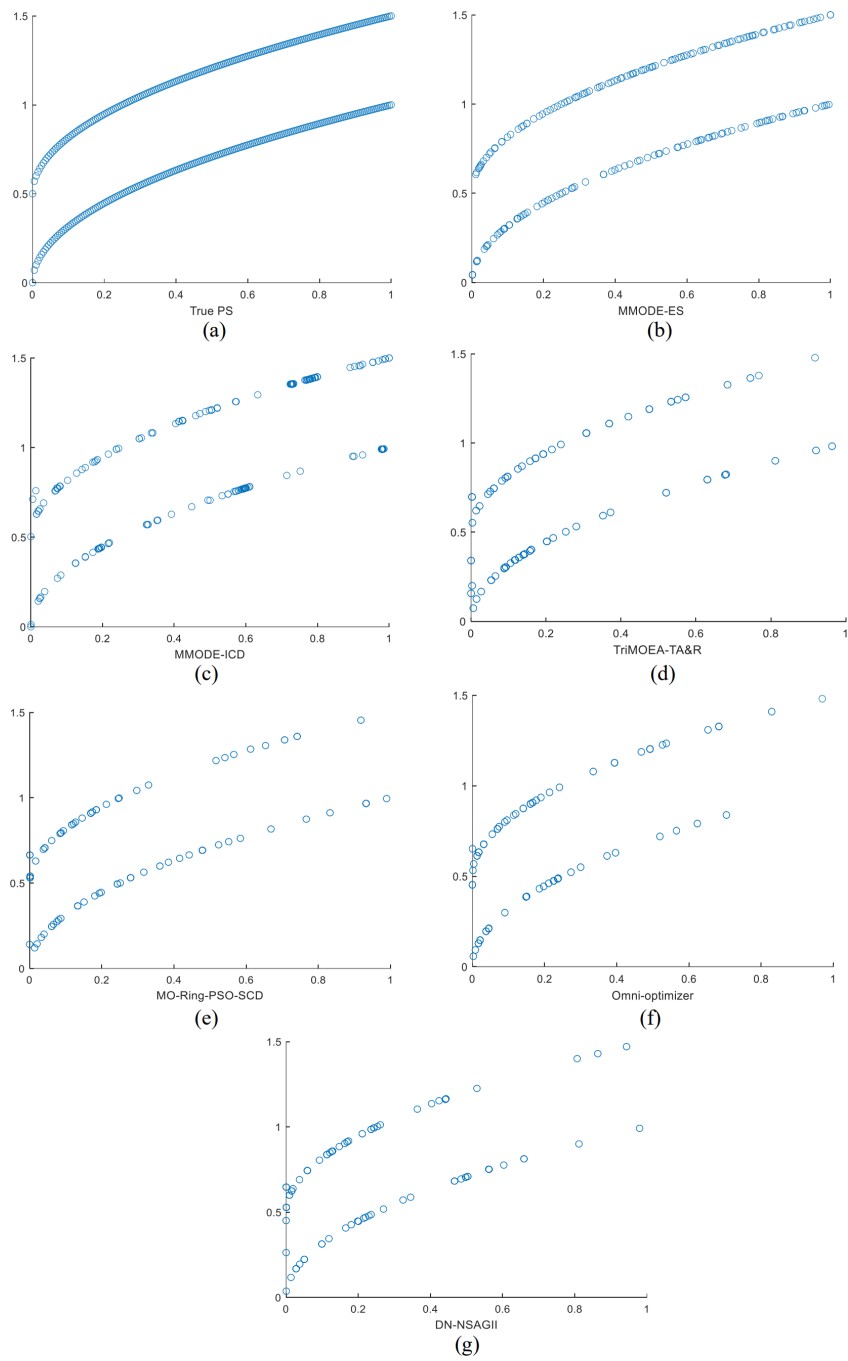

**Figure 1** The results of the decision space for each algorithm on the MMF3 function.

the performance of the algorithm in the objective space, but the consideration is not comprehensive enough. In contrast, MMODE_ES is an excellent algorithm for solving MMOPs

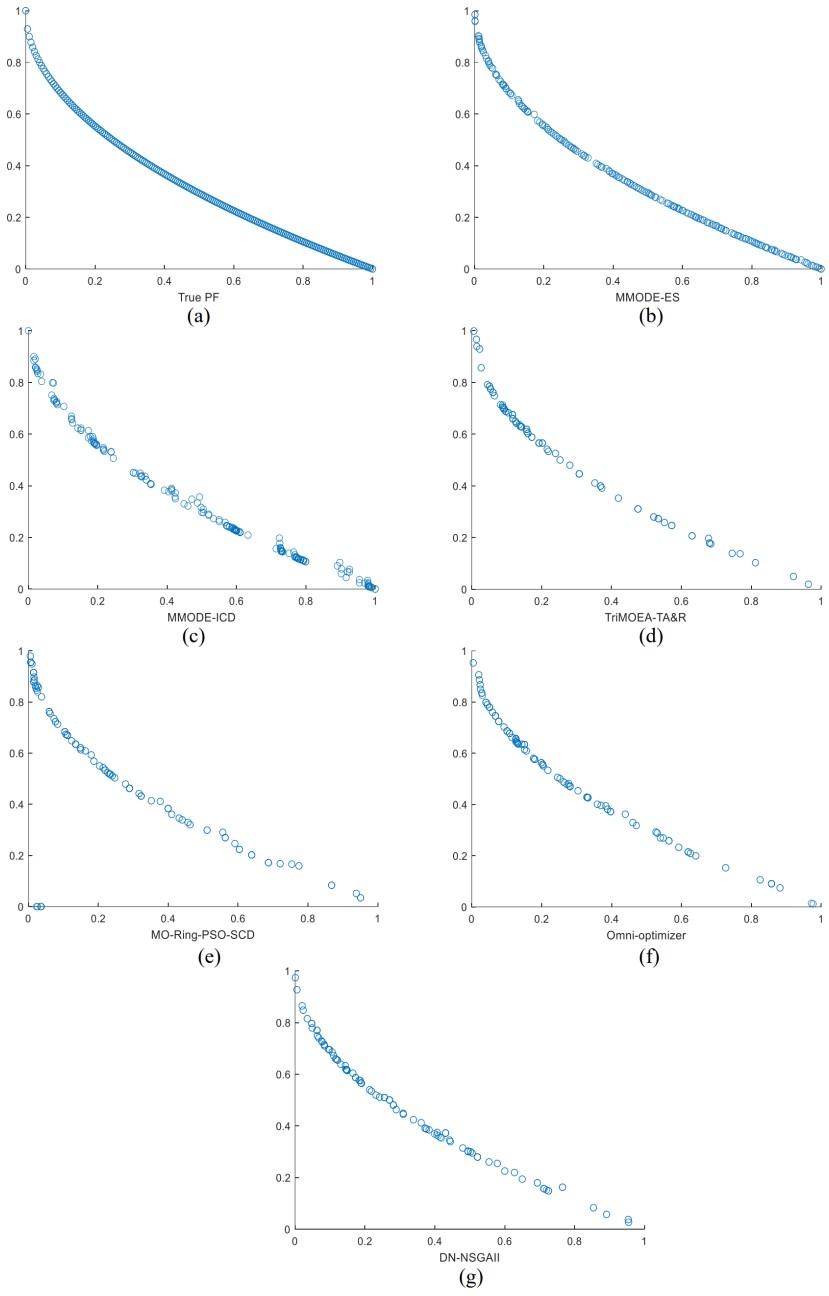

**Figure 2** **(A–G) The results of the objective space for each algorithm on the MMF3 function.**

## CONCLUSION

The article proposes a DE algorithm with hierarchical selection, which improve the quality of the PSs and the PF, when solving the MMOPs. Firstly, the MMODE_ES in this article designs two variational strategies based on special crowding distances for the degree of crowding of populations, which takes into account the diversity of populations, in the meantime to improve the distribution of solutions. In the stage of environment selection, a

stratification-based selection strategy is designed; the operation enables the elite individuals to be selected, and the algorithm can search in the direction of potential while letting the low-ranking individuals get the likelihood to explore; environment selection improves the diversity and convergence of the PSs and PF. Finally, an adaptive differential variation strategy is performed for the individuals after the non-dominated ranking has been obtained according to their convergence and diversity, which improves the convergence of the population if its variety is good and improves its diversity if the convergence is maintained. These strategies help the algorithm MMODE_ES to obtain more diverse and uniformly distributed PSs and PF, which improves the convergence of MMODE_ES. Finally, MMODE_ES is compared with several excellent MMO algorithms on the CEC2019, and results in the the *rPSP* and *rHV* show that MMODE_ES in this article is effective for solving the MMOPs.

Of course, the MMODE_ES proposed still has some shortcomings. First, MMODE_ES needs to perform three stages of evolution, thus leading to relatively high computational complexity and longer running time. Second, the performance of MMODE_ES in the decision space and the objective space needs to be further improved. Third, individual diversity and convergence in the evolution process need further research.

In future work, firstly, improving the deficiencies of MMODE_ES. Then, it is vital to improve further the diversity and convergence, and the adaptive parameter mechanism can be considered to enhance the searchability of the algorithm. At the end, considering applying it to more sophisticated and practical problems.

### Funding

This work was supported by the National Natural Science Foundation of China (No. 61862051 and No. 62241206), the Science and Technology Plan Project of Guizhou Province (No. (ZK[2022]449 and No. ZK[2022]550) and the program of Qiannan Normal University for Nationalities (No.2024zdzk03). There was no additional external funding received for this study. The funders had no role in study design, data collection and analysis, decision to publish, or preparation of the manuscript.

### Grant Disclosures

The following grant information was disclosed by the authors:
The National Natural Science Foundation of China: No. 61862051, No. 62241206.
The Science and Technology Plan Project of Guizhou Province: No. ZK[2022]449, No. ZK[2022]550.
The program of Qiannan Normal University for Nationalities:  No. 2024zdzk03.

### Competing Interests

The authors declare there are no competing interests.

## Author Contributions

- Xiao Wang conceived and designed the experiments, performed the experiments, analyzed the data, performed the computation work, prepared figures and/or tables, authored or reviewed drafts of the article, and approved the final draft.
- Dan Wang conceived and designed the experiments, authored or reviewed drafts of the article, and approved the final draft.
- Jincheng Zhou conceived and designed the experiments, authored or reviewed drafts of the article, and approved the final draft.

## Data Availability

The code and CEC2019 test set are available in the Supplemental Files.

## Supplemental Information

Supplemental information for this article can be found online at http://dx.doi.org/10.7717/peerj-cs.2182#supplemental-information.

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
