# Peer review of "Multimodal multi-objective optimization algorithm based on hierarchical environment selection strategy"

_PeerJ Computer Science, doi:10.7717/peerj-cs.2182_

## Round 0.1 · original submission · Major Revisions

Dear authors,

Thank you for submitting your article. Feedback from the reviewers is now available. It is not recommended that your article be published in its current format. However, we strongly recommend that you address the issues raised by the reviewers, especially those related to readability, experimental design and validity, and resubmit your paper after making the necessary changes.

Best wishes,

Reviewer 1 ·

Basic reporting

The paper presents an optimization algorithm, MMODE_ES, designed to address the limitations of existing multimodal multi-objective optimization (MMO) algorithms in achieving comprehensive and convergent Pareto solution sets. It introduces a hierarchical environment selection strategy alongside a use of crowding distance for individual variation, promoting diversity within populations. This approach utilizes non-dominated sorting coupled with a layered individual selection process based on an adaptive ratio, enhancing the potential for discovering intermediary, convergent, and diverse non-dominated sorted populations. The algorithm employs various differential reproduction strategies to avoid premature convergence, thereby maintaining robust search capabilities. This paper also compares with multiple MMO algorithms to compare the effectiveness. There are several problems I would like to mention:

1.In the introduction section, the authors list too many related works. I may recommend putting some related works into another related work section(consider creating a new section named related work).
2.Add some motivation about the algorithm proposed by this paper into the introduction section. Currently, the authors use only one paragraph to illustrate their approach in the introduction section, which is not enough. The authors may consider discussing at a high level, how the proposed algorithm can solve the issues of prior works.
3.For ‘PS’, use the full name instead of the abbreviation the first time it appears. Add introduction to PS in the Background section.
4.This paper needs to improve writing. e.g. lines 293-296, ‘To illustrate MMODE_ESís effectiveness in this paper, five excellent MMO algorithms are selected for comparison test, which are MO_Ring_PSO_SCD (Yue et al.,2018), DN-NSGAII (Liang et al.,2016), Omni-optimizer (Deb et al.,2005), TriMOEA-TA&R (Liu et al.,2018), MMODE_ICD (Yue et al.,2021) these algorithms.’ It is quite strange to use ‘these algorithms’ at the end of this sentence.
5.formatting issue, use link for doi reference.

Experimental design

Algorithm 1 and 2, use more mathematical forms or, fewer sentences: see algorithm definition in MO_Ring_PSO_SCD (Yue et al.,2018)

Validity of the findings

Consider using mean ± std dev for the tables 1 and 2.
The authors may consider adding experiments on different population sizes.

Reviewer 2 ·

Basic reporting

The manuscript contains novel elements. However, it presents some aspects that need to be solved before reconsideration.
The authors should explicitly mention the significant contributions of the manuscript. The novelty of the paper is not highlighted.
In the introduction section, add some other interesting multimodal algorithms such as “Evolutionary-Mean shift algorithm for dynamic multimodal function optimization” and “A competitive memory paradigm for multimodal optimization driven by clustering and chaos”
The advantages and limitations of the proposed approach in relationship with similar schemes is not clear.
Please revise the structure of the paper. It is recommendable to add in each section a couple of sentences that explain which is the purpose of the section. With this organization, the reader can clearly understand the sequence of the paper.

Experimental design

The experimental design adheres to established scientific standards, ensuring its robustness and reliability. Consequently, there are no comments in this section as the methodology employed meets the requisite criteria for accuracy and reproducibility in scientific research. This compliance with scientific norms underscores the credibility of the experimental results presented.

Validity of the findings

The validity of the results is significant due to their derivation from the scientific method. This methodological approach ensures that the findings are both reliable and reproducible, providing a sound basis for conclusions. By adhering to rigorous scientific protocols, the study establishes a strong foundation for the credibility and significance of its outcomes.

Additional comments

No comments

Reviewer 3 ·

Basic reporting

1- Proofreading is required.
2- The considered literature limitations are lacking.
3- The authors did not fully adhere to the journal's template instructions.

Experimental design

The manuscript falls within the scope of the journal.
The clarity of the test functions is lacking.

Validity of the findings

The reviewer appeared to have difficulty grasping the fundamental contributions of the manuscript.
Numerous abbreviations were used without prior definition, leading to potential confusion.
The discussion of the results lacked efficiency and clarity.
The conclusion seems to inadequately reflect the depth of the work, as perceived by the reviewer.

Annotated reviews are not available for download in order to protect the identity of reviewers who chose to remain anonymous.

---

## Round 0.2 · Minor Revisions

Dear authors,

Thank you for submitting your revised article. Your article has still not been recommended for publication in its current form. However, we do encourage you to address the concerns and criticisms of the reviewers and resubmit your article once you have updated it accordingly. It will be better to address the following for the quality:

1. The paper lacks the running environment, including software and hardware. The analysis and configurations of experiments should be presented in detail for reproducibility. It is convenient for other researchers to redo your experiments and this makes your work easy acceptance. A table with parameter settings for experimental results and analysis should be included in order to clearly describe them.
2. You should clarify the pros and cons of the methods. What are the limitation(s) methodology(ies) adopted in this work? Please indicate practical advantages, and discuss research limitations.
3. The research gaps and contributions should be clearly summarized in the introduction section. Please evaluate how your study is different from others in the related work section.
4. Equations should be polished. Explanation of the equations should be checked. Use "equation" not formula. Both "equation" and "formula" is used in the text. All variables should be written in italic as in the equations. Their definitions and boundaries should be explained. Relevant references should be given for the equations.

Best wishes,

Reviewer 3 ·

Basic reporting

Minor proofreading is required, based on the notes in the attached file.

Experimental design

The authors fulfill the required.

Validity of the findings

The authors fulfill the required.

Additional comments

Dear Authors,

Please find attached the files containing specific notes and recommendations for your manuscript. Kindly review and incorporate these suggestions to enhance the quality of your work.

Sincerely,

Annotated reviews are not available for download in order to protect the identity of reviewers who chose to remain anonymous.

---

## Round 0.3 · accepted · Accept

Dear authors,

Thank you for the revision and for clearly addressing all the reviewers' comments. I confirm that the paper is improved. Your paper is now acceptable for publication in light of this revision.

Best wishes,

Reviewer 3 ·

Basic reporting

No further revision is required.

Experimental design

No further revision is required.

Validity of the findings

No further revision is required.

Additional comments

No further revision is required.